# Micro-Computed Tomography Analysis of Peri-Implant Bone Defects Exposed to a Peri-Implantitis Microcosm, with and without Bone Substitute, in a Rabbit Model: A Pilot Study

**DOI:** 10.3390/bioengineering11040397

**Published:** 2024-04-19

**Authors:** Camila Panes, Iván Valdivia-Gandur, Carlos Veuthey, Vanessa Sousa, Mariano del Sol, Víctor Beltrán

**Affiliations:** 1Center of Excellence in Morphological and Surgical Studies (CEMyQ), School of Medicine, University of La Frontera, Temuco 4780000, Chile; camila.panes@ufrontera.cl (C.P.); carlos.veuthey@ufrontera.cl (C.V.); mariano.delsol@ufrontera.cl (M.d.S.); 2PhD Program in Morphological Sciences, Universidad de La Frontera, Temuco 4780000, Chile; 3Faculty of Dentistry, Universidad de La Frontera, Temuco 4780000, Chile; 4Biomedical Department, Universidad de Antofagasta, Antofagasta 1270300, Chile; 5Periodontology and Periodontal Medicine, Centre for Host-Microbiome Interactions, Faculty of Dentistry, Oral and Craniofacial Sciences, Kings College London, Guy’s and St Thomas’ NHS Foundation Trust, London SE1 9RT, UK; vanessa.sousa@kcl.ac.uk; 6Clinical Investigation and Dental Innovation Center, Dental School and Center for Translational Medicine, Universidad de La Frontera, Temuco 4780000, Chile

**Keywords:** peri-implant bone defects, peri-implantitis, critical size defects, animal model, microcomputed tomography, dental implant

## Abstract

Peri-implantitis is an inflammatory condition characterized by inflammation in the peri-implant connective tissue and a progressive loss of supporting bone; it is commonly associated with the presence of biofilms on the surface of the implant, which is an important factor in the development and progression of the disease. The objective of this study was to evaluate, using micro-CT, the bone regeneration of surgically created peri-implant defects exposed to a microcosm of peri-implantitis. Twenty-three adult New Zealand white rabbits were included in the study. Bone defects of 7 mm diameter were created in both tibiae, and a cap-shaped titanium device was placed in the center, counter-implanted with a peri-implantitis microcosm. The bone defects received a bone substitute and/or a resorbable synthetic PLGA membrane, according to random distribution. Euthanasia was performed 15 and 30 days postoperatively. Micro-CT was performed on all samples to quantify bone regeneration parameters. Bone regeneration of critical defects occurred in all experimental groups, with a significantly greater increase in cases that received bone graft treatment (*p* < 0.0001), in all measured parameters, at 15 and 30 days. No significant differences were observed in the different bone neoformation parameters between the groups that did not receive bone grafts (*p* > 0.05). In this experimental model, the presence of peri-implantitis microcosms was not a determining factor in the bone volume parameter, both in the groups that received regenerative treatment and in those that did not.

## 1. Introduction

In the latest International Workshop for the Classification of Periodontal and Peri-implant Diseases and Conditions (2017) [1], peri-implantitis (PI) was defined as a pathological condition associated with bacterial biofilm occurring in the tissues around dental implants, characterized by inflammation of the peri-implant mucosa and a progressive loss of supporting tissues. For the diagnosis of this pathology, peri-implant bone loss measured radiographically is evaluated and added to an increase in probing depth and bleeding and/or suppuration when examining the tissues around the implant [2]. The aim of PI treatment is to establish inflammation-free peri-implant soft tissues, eliminate all biofilm retention factors and prevent further bone loss. To date, the therapies recommended for the treatment of this pathology include, in addition to the elimination of etiological factors, different methods of decontamination and peri-implant regenerative therapies. Precisely, the definition of the bone defect and the volume of peri-implant bone lost are factors that condition the choice of regenerative therapy, in which the use of resorbable membranes, bone grafts or substitutes and guided bone regeneration obtain results that, although promising, do not guarantee the success of the infected implant [3].

Since the use of animal models remains relevant in the study of bone pathologies [4], the study of peri-implantitis has been explored in several animal models, mostly large animals such as dogs, non-human primates, pigs and sheep, as well as small animals such as rats and rabbits [5]. Dogs, which have a natural susceptibility to periodontitis and a similar bone composition to humans, have proven to be a successful model for the simulation of peri-implant disease using a ligation model, which is considered the gold standard for the induction of peri-implant diseases in animal models [6]. The ligation model allows a simulation of mucositis and PI very similar to the natural progression of peri-implant disease since it enhances the accumulation of biofilm and allows the free progression of inflammation and bone defects; however, the animal under study requires several surgical interventions, in addition to longer periods of exposure. On the other hand, although non-human primates have the same advantages as a model for the study of oral diseases, they also have important restrictions to their use, such as the regulations of each country for their acquisition and maintenance and the need to use small-sized implants, in addition to the ethical considerations involved in the development of research in these animals. In the case of experimental models with miniature pigs, their bone similarity to human bone in both composition and alveolar remodeling can be highlighted [7], and they are frequently used in implantological research due to their similarity in the osseointegration process and as a model of the impact of systemic diseases in this process [8]. In addition to this, and like the dog model, they have the advantage of using the oral environment without the need for small implants [5].

On the other hand, experimental models in small animals also allow the use of the ligation model, and infra- and supracrestal bone defects can be obtained in rodents, with circumferential marginal bone loss, consistent with the clinical presentation of PI [9], in addition to their advantages related to lower costs and ease of handling and housing allowing the use of a greater number of specimens. However, the approaches are mainly extraoral, due to the difficulty in accessing the oral cavity because of its reduced dimensions and low opening range, using surgical procedures that acutely generate a peri-implant bone defect. Precisely, the experimental rabbit model is used in different areas of interest, such as the study of biomaterials in both regenerative medicine [10,11,12] and dental implants [13,14]. For the study of peri-implant bone regeneration in cases of implants affected by PI, in vivo models have previously been used in New Zealand rabbit (*Oryctolagus cuniculus*) tibiae because this allows the analysis of peri-implant bone defects and regenerative procedures associated with their treatment [15,16].

Although the implant insertion site in the tibia lacks an infectious microbiome per se, as reported by Sousa et al. [17], it is possible to use caps previously contaminated with certain bacterial microcosms present in PI (including *Streptococcus* spp., *Actinomyces* spp., *Staphylococcus* spp., *Candida* spp., *Veillonella* spp.), which make it possible to evaluate the antibacterial capacity of the biomaterials used and the osteogenic repair achieved in peri-implant defects over different periods of time.

The aim of this study was to evaluate, by means of a pilot model, the bone regeneration of peri-implant bone defects of known critical size exposed to titanium caps contaminated with a bacterial PI microcosm in New Zealand rabbit (*Oryctolagus cuniculus*) tibiae and quantify bone regeneration parameters in peri-implant bone defects exposed to a PI microcosm in the presence or absence of biomaterials.

## 2. Materials and Methods

### 2.1. Study Design

This study was approved by the Scientific Ethical Committee of the Universidad de La Frontera, Temuco, Chile (folio 106/20). It included 23 New Zealand white rabbits (*Oryctolagus cuniculus*), weighing approximately 3.5 kg; they were reared and maintained in the biotherium of the Center of Excellence in Morphological and Surgical Studies (CEMyQ), with a 12 h light and dark regime and food and water ad libitum. The intervention of the animals was performed according to the corresponding ethical guidelines, the animal monitoring protocol created by Morton and Griffiths (1985) [18] and the ARRIVE recommendations for animal experimentation [19]. The 23 rabbits were divided into groups as shown in Figure 1. From each rabbit, both tibiae were treated and assigned to a study group randomly. Each tibia received a synthetic bone substitute based on β-tricalcium phosphate and hydroxyapatite (Calc-i-oss Crystal+, GUIDOR) and/or synthetic resorbable PLGA membrane (ES membrane, GUIDOR), as shown in Figure 1, leaving a negative control group (without the use of biomaterials) and a positive control group (leaving the cap contaminated with the microcosm, without going through the disinfection process; method explained in “surgical procedures”).

### 2.2. Contamination of Titanium Caps

Titanium caps 4.0 mm in diameter and 2.5 mm high (JDentalCare Implant, Italy), were designed to be inserted on the head of each implant. These caps were subjected to a contamination process with a PI microcosm on their external surface, following the model of Sousa et al. [17]; from a saliva suspension collected from 20 healthy patients over 18 years of age, biofilm formation was induced on the external surface of the caps, contained in 6-well plates (NunclonTM Delta Surface 6-well plates, Nunc, Denmark). These plates were incubated at 37 °C for 14 days, in a 5% CO_2_-supplemented atmosphere, with a culture medium consisting of RPMI-1640 (+25 mM HEPES, + L-glutamine, Cytiva HyCloneTM, UT, USA) and equine serum (Cytiva, HyCloneTM UT, EE.UU) in a 3:2 ratio. To each 100 mL of the RPMI mixture with equine serum, 5 µL of menadione stock (10 mg/mL) and 100 µL of hemin stock (5 mg/mL) were added. The medium was replaced every 48 h, according to color and turbidity.

### 2.3. Surgical Procedures

All surgical procedures were performed after the animal was weighed, followed by a pre-anesthetic dose of intramuscular buprenorphine and anesthesia (35 mg/kg ketamine supplemented with 5 mg/kg xylazine intramuscularly). A single surgical intervention was performed per animal, on both tibiae. After the skin was shaved and the area to be treated was disinfected with 0.12% chlorhexidine, dissection by planes was started in the proximal tibia area; circular bone defects were created using a 7 mm diameter surgical trephine at low speed and irrigation with sterile 0.9% sodium chloride solution, in order to obtain bone defects of known diameter, in the median area of the diaphysis, under the epiphyseal line. A prefabricated titanium implant 3.7 mm in diameter and 10 mm in length (JDentalCare Implant, Modena, Italy), was inserted in each tibia in the center of the created bone defect. Subsequently, the implant was covered with the cap. Just before insertion and according to the randomly assigned study group, the biofilm-contaminated cap was decontaminated by mechanical brushing treatment with titanium fibers (RotoBrush-Titanium, SALVIN, Charlotte, NC, USA) and photodynamic therapy with λ 810 nm diode laser NV PRO3 Microlaser INTL 808NM (DenMat, Lompoc, CA, USA); the caps were impregnated with a 2% methylene blue solution to generate the cavitation phenomenon, and without direct contact with the cap surface, the laser was applied in continuous mode at 1.0 W power.

To minimize the risk of infection in the animals, injectable antibiotics and anti-inflammatory drugs were administered for 3 days after surgery: enrofloxacin (5 mg/kg; Vetoquinol, Biowet) and meloxicam (0.4 mg/kg; Meloven, Dopharma). At 15 and 30 days post-surgery, the rabbits were euthanized by sodium pentobarbital at 200 mg/kg body weight. Then, both tibiae were extracted, cut at the diaphysis with a saw and stored in 4% formalin diluted in phosphate buffer (0.1 M and pH 7.4) in a volume of 1:10. Afterward, the samples were reduced to the site closest to the bone defect, in order to obtain samples approximately 1.5 cm thick, and stored in 4% buffered formalin until further analysis.

### 2.4. Micro-CT Imaging Study

A micro-CT scanner (SkyScan 1278, Bruker, Kontich, Belgium) with the following parameters was used for quantification of neoformed bone in all samples: camera pixel size of 75 μm, 59 kV X-ray tube power, 692 μA X-ray beam intensity, voxel size of 51 μm^3^ and 1 mm thick aluminum filter. The acquired images were reconstructed into a 3D dataset using NRecon v.1.6.9 software (Bruker, Kontich, Belgium). The images were spatially reoriented following the axial axis of the implant (Figure 2) using DataViewer v.1.5.6.2 software (Bruker-microCT, Kontich, Belgium), and segmentation of the structures was performed using CTAn v.1.12 software (Bruker, Kontich, Belgium).

Considering the dimensions of the titanium cap and the peri-implant bone defect generated, the volume of interest (VOI) was established as a cylinder with a circumference of 7 mm in diameter, corresponding to the bone defect generated in surgery, and 4 mm in height to cover the entire surface in contact with the titanium cap. Basic morphometric indices consider the measurement of total volume of interest (TV), bone volume (BV) and bone surface (BS), with ratios between these parameters providing information about bone structure [20]. For this study, the values of mineralized tissue contained in the VOI were measured using 5 parameters: bone volume (BV), percent bone volume (BV/TV), bone surface (BS), bone surface/volume ratio (BS/BV) and bone surface density (BS/TV).

### 2.5. Statistical Analysis

The SPSS v.24.0 program (IBM Corp., Armonk, NY, USA) was used for statistical analysis. The data were presented as mean ± standard deviation. The normality of the data generated by the microtomographic analyses was examined using the Shapiro–Wilk test. Differences between the different materials at 15 and 30 days were determined by one-way analysis of variance (ANOVA) and its non-parametric variable, and they were complemented with Tukey’s test. The Student’s *t*-test for unpaired samples and its non-parametric test (Kruskal–Wallis) were used to determine differences between materials at 15 and 30 days. A *p*-value < 0.05 was considered significant.

## 3. Results

The parameters obtained from the analysis of the microtomographic dataset are summarized in Table 1 and Table 2, and the significant differences observed are plotted in Figure 3 and Figure 4, while the comparison between both post-surgical periods is in Table 3.

### 3.1. Bone Volume (BV) and Percent Bone Volume (BV/TV)

The value of bone volume and the percentage of bone measured within the VOI showed significant differences between the groups that received bone grafts and those that did not (*p* < 0.0001). On the other hand, there were no significant differences between the groups that received bone grafts, associated or not with the PLGA membrane, at 15 and 30 days post-surgery.

Regarding the volume of bone obtained, although there was an increase in mm^3^ of neoformed tissue between 15 and 30 days post-surgery, no significant differences were observed between the two time periods, except in the group that received bone grafts alone, in which a decrease in new bone was observed within the VOI (*p* < 0.05).

### 3.2. Bone Surface (BS)

The bone surface area value at 15 days showed significant differences between the groups that received bone grafts and those that did not (*p* < 0.0001). On the other hand, there were no significant differences between the groups that received bone grafts, either with or without the presence of the PLGA membrane. On the other hand, at 30 days, the group that received both biomaterials only had significant differences with the group that had the contaminated surface of the cap.

When comparing the bone surface obtained between 15 and 30 days, significant differences were observed for the tibiae that received bone grafting together with the PLGA membrane, a period in which the bone surface quantified decreased (*p* < 0.05).

### 3.3. Bone Surface Density (BS/TV)

The bone surface density of the group that received both biomaterials decreased between 15 and 30 days (*p* < 0.05), showing significant differences with all the other groups at 15 days. In the evolution of this parameter, at 30 days, the differences were concentrated towards the groups that had not received biomaterials (*p* < 0.01).
bioengineering-11-00397-t001_Table 1Table 1Mean and standard deviation of the parameters quantified in the 15-day experimental groups.
PLGAMBBGCRTUCBV (mm^3^)10.61 ± 1.6040.32 ± 6.4955.00 ± 1.3210.40 ± 1.7313.18 ± 2.86BV/TV (%)11.69 ± 1.7744.44 ± 7.1660.62 ± 1.4611.46 ± 1.9114.53 ± 3.15BS (mm^2^)109.80 ± 9.17181.10 ± 18.26166.40 ± 10.52110.70 ± 7.17131.20 ± 13.30BS/BV (^1^/_mm_)10.47 ± 1.014.53 ± 0.463.03 ± 0.2310.82 ± 1.3310.16 ± 1.23BS/TV (^1^/_mm_)1.20 ± 0.101.99 ± 0.191.83 ± 0.111.22 ± 0.071.44 ± 0.14BV: bone volume; BV/TV: percent bone volume; BS: bone surface; BS/BV: bone surface/volume ratio; BS/TV: bone surface density; PLGA: PLGA membrane; MB: membrane and bone graft; BG: bone graft; CRT: control group; UC: untreated cap.
Figure 3Graphs of the parameters obtained by computerized microtomography, 15 days post-surgery: (**A**) Bone volume (BV) present in the VOI. (**B**) Percent bone volume (BV/TV). (**C**) Bone surface (BS). (**D**) Bone surface/volume ratio (BS/BV). (**E**) Bone surface density (BS/TV). ** *p* ≤ 0.001; *** *p* ≤ 0.0001. PLGA: PLGA membrane; MB: membrane and bone graft; BG: bone graft; CRT: control group; UC: untreated cap.
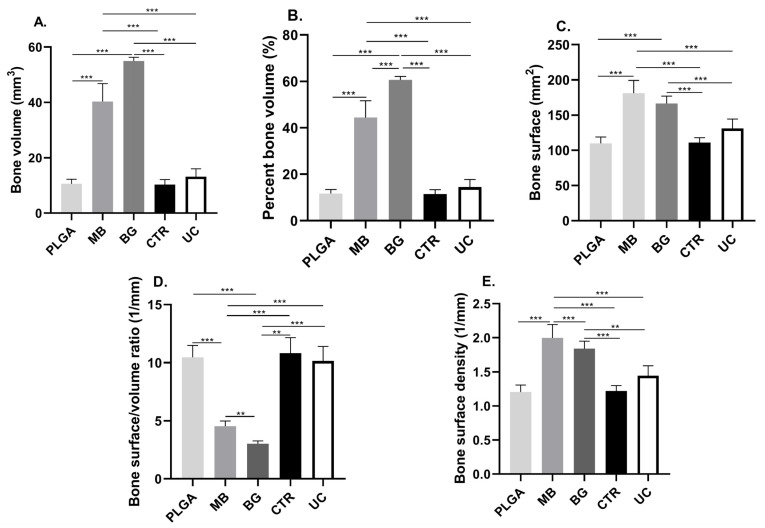

bioengineering-11-00397-t002_Table 2Table 2Mean and standard deviation of the parameters quantified in the 30-day experimental groups.
PLGAMBBGCRTUCBV (mm^3^)11.01 ± 2.3948.38 ± 4.8349.29 ± 6.2313.83 ± 6.009.66 ± 2.39BV/TV (%)12.14 ± 2.6353.33 ± 5.3254.33 ± 6.8615.25 ± 6.6110.65 ± 2.64BS (mm^2^)119.6 ± 20.86149.80 ± 10.02167.8 ± 12.95124.90 ± 25.74102.60 ± 19.49BS/BV (^1^/_mm_)10.98 ± 0.803.12 ± 0.393.46 ± 0.619.83 ± 2.4910.75 ± 0.97BS/TV (^1^/_mm_)1.31 ± 0.231.65 ± 0.101.85 ± 0.141.37 ± 0.281.13 ± 0.21BV: bone volume; BV/TV: percent bone volume; BS: bone surface; BS/BV: bone surface/volume ratio; BS/TV: bone surface density; PLGA: PLGA membrane; MB: membrane and bone graft; BG: bone graft; CRT: control group; UC: untreated cap.
Figure 4Graphs of the parameters obtained by computerized microtomography, 30 days post-surgery: (**A**) Bone volume (BV) present in the VOI. (**B**) Percent bone volume (BV/TV). (**C**) Bone surface (BS). (**D**) Bone surface/volume ratio (BS/BV). (**E**) Bone surface density (BS/TV). * *p* ≤ 0.01; ** *p* ≤ 0.001; *** *p* ≤ 0.0001. PLGA: PLGA membrane; MB: membrane and bone graft; BG: bone graft; CRT: control group; UC: untreated cap.
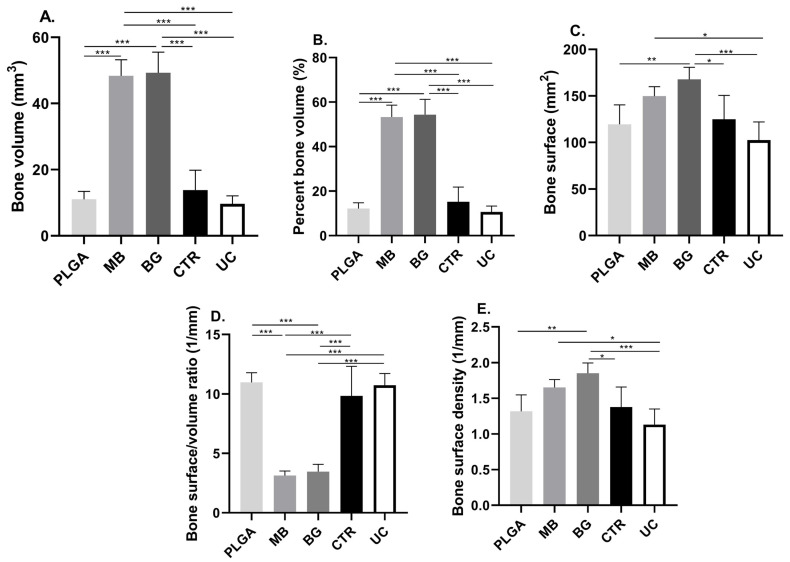

bioengineering-11-00397-t003_Table 3Table 3Comparison of microtomographic parameters measured for all experimental groups, at 15 and 30 days post-surgery.
15 Days30 Days**PLGA membrane**

BV (mm^3^)10.61 ± 1.60t11.01 ± 2.39BV/TV (%)11.69 ± 1.77t12.14 ± 2.63BS (mm^2^)109.80 ± 9.17t119.6 ± 20.86BS/BV (^1^/_mm_)10.47 ± 1.01mw10.98 ± 0.80BS/TV (^1^/_mm_)1.20 ± 0.10t1.31 ± 0.23**Membrane and bone graft**

BV (mm^3^)40.32 ± 6.49t48.38 ± 4.83BV/TV (%)44.44 ± 7.16t53.33 ± 5.32BS (mm^2^)181.10 ± 18.26t149.80 ± 10.02BS/BV (^1^/_mm_)4.53 ± 0.46mw3.12 ± 0.39BS/TV (^1^/_mm_)1.99 ± 0.19t1.65 ± 0.10**Bone graft**

BV (mm^3^)55.00 ± 1.32t49.29 ± 6.23BV/TV (%)60.62 ± 1.46t54.33 ± 6.86BS (mm^2^)166.40 ± 10.52t167.8 ± 12.95BS/BV (^1^/_mm_)3.03 ± 0.23mw3.46 ± 0.61BS/TV (^1^/_mm_)1.83 ± 0.11t1.85 ± 0.14**Control group**

BV (mm^3^)10.40 ± 1.73t13.83 ± 6.00BV/TV (%)11.46 ± 1.91t15.25 ± 6.61BS (mm^2^)110.70 ± 7.17t124.90 ± 25.74BS/BV (^1^/_mm_)10.82 ± 1.33mw9.83 ± 2.49BS/TV (^1^/_mm_)1.22 ± 0.07t1.37 ± 0.28**Untreated cap**

BV (mm^3^)13.18 ± 2.86t9.66 ± 2.39BV/TV (%)14.53 ± 3.15t10.65 ± 2.64BS (mm^2^)131.20 ± 13.30t102.60 ± 19.49BS/BV (^1^/_mm_)10.16 ± 1.23mw10.75 ± 0.97BS/TV (^1^/_mm_)1.44 ± 0.14t1.13 ± 0.21t: Student’s *t*-test; mw: Mann–Whitney U test; BV: Bone volume; BV/TV: Percent bone volume. BS: Bone surface. BS/BV: Bone surface/volume ratio. BS/TV: Bone surface density. 


## 4. Discussion

Standardized bone defects, with a known extent and depth, represent an advantage over models that reproduce peri-implantitis resorption in the oral cavity, which generates bone defects of different magnitudes [21], over a longer period of time and with the need to subject animals to a greater number of surgical interventions [22]. This methodological advantage was also reflected in the process of data collection and analysis of the micro-CT study, for which the establishment of a reproducible and representative interest volume allowed an accurate comparison between the different experimental groups. In this study, although a low threshold voltage (close to 50 kV) was used in the micro-CT scanner, the use of a larger voxel size and the artifacts generated by the presence of the implant metal during image acquisition affected the morphometric and density measurements and, therefore, also the indices obtained. In general, small animals such as rodents and lagomorphs require voxel sizes between 20 and 60 µm [23,24], relative to the average trabecular structure of the model studied. However, the voxel size applied was much larger than those in other studies that have analyzed rabbit bone morphometry (14–19 µm) [25], although it was sufficient for trabecular observation according to Voor et al. [26] in this type of model. However, in order to perform an effective and accurate comparison between experimental models using micro-CT for bone microarchitecture assessment, factors specific to image acquisition and processing must be considered [27,28], which must be established in the methodology of the study.

Previous research using animal models for the study of peri-implant disease highlights that the dimensions of the bone defect, modifications in the implant surface and the use of biomaterials modify the response of the affected bone and potentially accelerate the regeneration process [16,29]. Although bone tissue has enormous regenerative potential, there are situations such as critical size defects, deficit of osteoinductive or nutritional factors, low blood supply and metabolic or hormonal factors that do not allow reparative osteogenesis with complete histotypic and organotypic recovery [29,30]. In this study, the critical bone defects generated were of equal or larger dimensions than those treated by other authors under similar conditions [16,31,32,33], although potentially at the limit according to others [34]. However, added to the uni-cortical bone defect generated, the influence of the implant insertion in the posterior cortex and the decontamination process of the titanium cap surfaces must be considered. Despite these variables, bone repair was evident in all experimental groups, independent of the biomaterial received and the disinfection treatment performed on the cap.

The use of resorbable membranes in critical bone defects is a strategy in guided bone regeneration, providing in the first instance a mechanical barrier that provides stability of the initial clot and of the grafted biomaterial as a bone substitute [35]. Since a second surgical intervention for the removal of non-resorbable membranes could be counterproductive for tissue regeneration, resorbable membranes are in widespread clinical use because they are biodegradable and avoid reoperation. These include membranes made from animal collagen and others from natural or synthetic polymers, such as PLA (polylactic acid), PGA (polyglycolic acid), PLC (polycaprolactone) and the copolymers PLGA (poly(lactic-co-glycolic acid)) and PLCL (poly(lactide-co-caprolactone)). PLGA membranes show good performance in their mechanical properties and biodegradability; some in vivo studies, such as that of Istumi et al. [36], have demonstrated their inhibitory action on cell proliferation and connective tissue invasion in bone defects, while modifications on their surface would allow promoting osteogenic proliferation and differentiation [37,38]. Although the benefits of using resorbable membranes can be enhanced by supplementing them with growth factors, nanoparticles or plasticizers as part of their composition [39], the lack of osteoconductive and osteoinductive factors in the membrane is not determinant in their function and biocompatibility. Precisely, in this study, the experimental groups that received resorbable membranes only showed significant differences in bone formation compared to the experimental groups that received bone grafts; the rest of the groups, which did not receive membranes or grafts, showed very similar results in terms of regenerated bone volume; therefore, the use of the membrane was not considered decisive in terms of the amount of neoformed bone. However, as stated by Delgado-Ruiz et al. [31], the use of a membrane should not be discarded and can be applied in a complementary way to bone grafting, since it is useful to control the disintegration or dispersion of particles between surgical planes, thus delimiting the area of the bone defect and avoiding the proliferation of unwanted soft tissue within it, together with the possible entry of contaminating materials in case of a premature loss of the superficial sutures, considering the often unpredictable environmental and behavioral conditions when using animal experimental models.

The results obtained here should be evaluated with caution since the particles of the bone graft used were also counted within the VOI given the methodological difficulty of extracting them from the analysis. Bone substitutes are biocompatible, osteogenic, osteoinductive and osteoconductive materials that undergo a resorption process in which they progressively decrease in volume as the formation of new bone tissue precedes it. Although autografts remain the gold standard for the reconstruction of bone defects, allografts are a viable alternative, especially in cases where extensive areas of bone loss need to be treated, such as in critical defects [40]. In this study, a fully synthetic, irregular granule, partially resorbable allograft was used to provide a long-term osteoconductive platform, which also allows for stability of the grafted site. This was evident from the observation that, in general, all experimental groups showed minimal increases in all parameters studied between 15 and 30 days. These results, similar to those reported by Trento et al. [16] and Delgado-Ruiz et al. [31], confirmed that bone substitutes increase bone density, although the analyses should possibly consider longer post-surgical times (2, 4 and 6 weeks) due to the subsequent remodeling process of the bone defects, and at longer observation times, the resorbable membranes can probably play a more important barrier role for avoiding the proliferation of non-bone tissues in the regeneration zone inside the PI bone defect. Despite methodological considerations that make it difficult to measure new bone without considering bone graft particles, 3D and high-precision imaging techniques such as micro-CT are considered the gold standard for the evaluation of bone morphology and microarchitecture in rodents and other small animals ex vivo [23,27], under normal, senescent or pathological conditions [20,41]. This technique allowed the simultaneous analysis of areas of interest considerably larger than what would have been possible to observe by conventional histology, and without the need to alter the sample, as is common in the study of calcified tissues. However, quantitative and qualitative histological techniques should not be discarded as they provide relevant information on cellularity and dynamic indices of bone remodeling [31,42]. Despite this, for the parameters measured in this study, and as indicated by Zenzes and Zaslansky [43], the correlation between these methodologies is high, both in the measurement of bone volume and the percentage of tissue occupied by neoformed bone within the defect generated.

The inflammatory response observed in the groups exposed to non-decontaminated caps should be further analyzed; in these groups, although the bone defects were not treated with any biomaterial, bone substitute or resorbable membrane, they had a significant reparative response to the injury, very similar to that observed in the groups treated without regenerative materials (PLGA membrane and/or decontaminated cap). This can be explained by the in situ presence of exposed bone marrow at the surgical site, as the presence of mesenchymal cells and abundant blood supply provided the osteoprogenitor cells necessary for regeneration [44], coupled with a surgical technique that stabilized the initial clot used as a platform in tissue regeneration. However, it must be taken into account that these results do not consider histological or histomorphometric analyses, through which the real effect of a regenerative therapy on a critical bone defect could be estimated, along with the quality of the bone achieved.

## 5. Conclusions

It is possible to use a bone regeneration model of critical defects exposed to a microcosm of peri-implantitis in a rabbit tibia due to its easy handling, the possibility of using a larger number of specimens, its rapid skeletal maturity, and the possibility of using standard-size implants in more than one surgical site. In this experimental model simulating peri-implant bone defects, the presence of a peri-implantitis microcosm was not determinant in the regeneration of bone volume, compared to the other groups that did not receive regenerative treatment, with similar results between them at 15 and 30 days, and with the presentation of minimal postoperative complications.

## Figures and Tables

**Figure 1 bioengineering-11-00397-f001:**
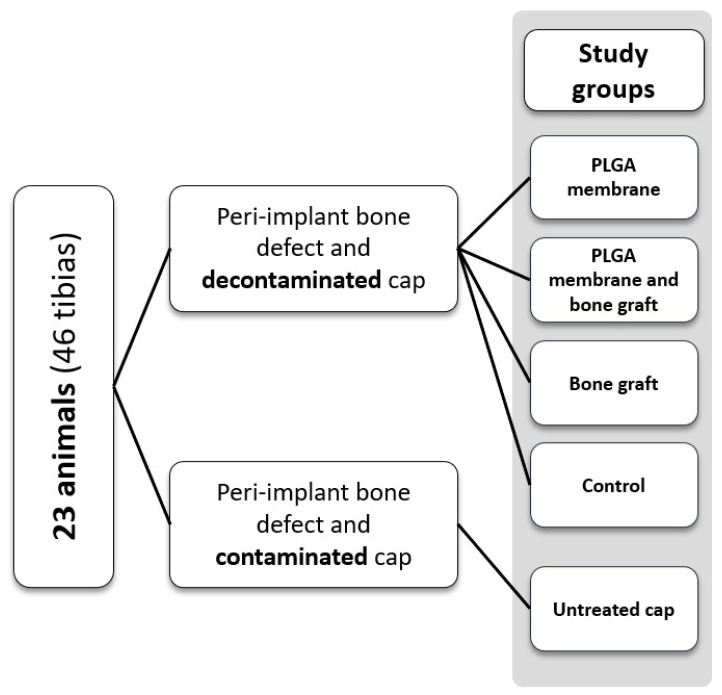
Schematization of the division of the experimental groups.

**Figure 2 bioengineering-11-00397-f002:**
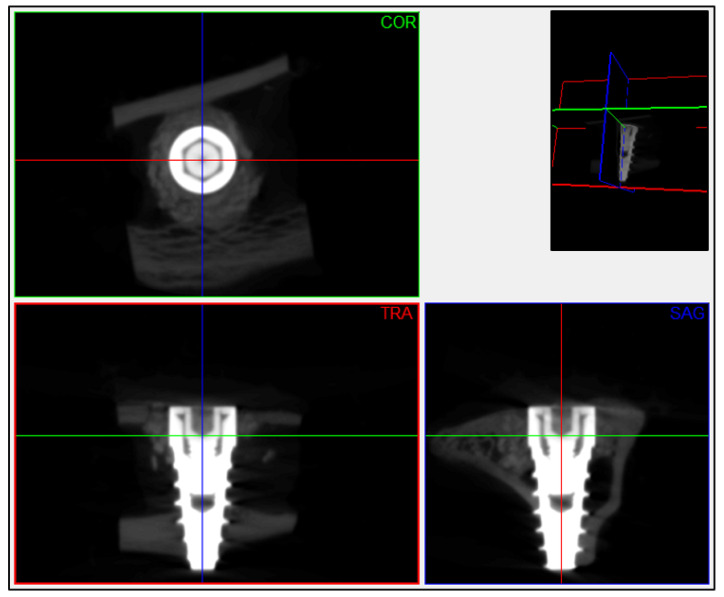
Example of the spatial reorientation applied to each scanned sample using DataViewer v.1.5.6.2 software (Bruker-microCT, Kontich, Bélgica), matching the axial axis and center of the implant. The images oriented in the coronal plane were extracted for subsequent segmentation and analysis of the images. COR: coronal plane; TRA: transverse plane; SAG: sagittal plane.

## Data Availability

The data presented in this study are available upon request from the corresponding author due to pending patent permissions for being a utility model.

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
