# Peer review of "Micro-Computed Tomography Analysis of Peri-Implant Bone Defects Exposed to a Peri-Implantitis Microcosm, with and without Bone Substitute, in a Rabbit Model: A Pilot Study"

_bioengineering, 2024, doi:10.3390/bioengineering11040397_

Round 1

Reviewer 1 Report

Comments and Suggestions for Authors

Abstract:

The authors have started their abstract by going straight to their main goals and aims. I advise the authors to give a short background on their topic first. Other than that, the overall tone of the abstract is appropriate and the main details of the methods and results have been reported.

Keywords:

I prefer changing "critical sized defects" to "critical sized bone defects".

Introduction:

I personally think the ASA 6/7 format of referencing is not appropriate for this kind of study. I suggest the authors change their referencing style into numbered like vancouver.

The introduction has a pleasing flow. I can see the authors have done the best they could in trying to convince the readers that this kind of study was needed to be part of the literature. The references they have chosen are all related to their subject. Most of the references are relatively newly-published. I respect the fact that the authors have chosen to run this pilot study before executing a full-on in vivo study. 

The methods and results section are both filled with all the key details in an organized manner. I highly respect the methodology behind this study. And from my own extensive electronic search I can tell that the originality of this study and the necessity of its existence in the literature are both on a high level of reliability. 

Reviewer 2 Report

Comments and Suggestions for Authors

In this article, the authors evaluated by means of micro-CT the bone regeneration of peri-implant defects surgically created and exposed to a microcosm of peri-implantitis. Bone defects of 7 mm in diameter were created in both tibiae and a cap-like titanium device was placed in the center of them, counter-implanted with a microcosm of peri-implantitis. The bone defects received a bone substitute and/or PLGA synthetic resorbable membrane, according to random distribution. In this experimental model, the presence of peri-implantitis microcosm was not determinant in bone volume parameter, both in the groups that received regenerative treatment and in those that did not. However, the paper needs minor improvement before acceptance for publication. My detailed comments are listed as follows:

1.    Artifacts arising from the presence of implanted metals during image acquisition affect morphometry and densitometry, and how to control this effect the authors need further clarification.

2.    The inflammatory response observed in the groups exposed to non-decontaminated caps should be further analyzed.

3.    The authors should have added about histological or histomorphometric analysis so that they could better compare the actual effects of regenerative treatments on bone defects

4.    Since the particles of the bone graft used are also counted within the VOI given the methodological difficulty of extracting them from the analysis, how should authors control the error caused by this factor?

5.    The influence of the implant insertion in the posterior cortex and the decontamination process of the titanium cap surfaces must be considered.

6.    Some paper could be cited in this paper: Engineered Regeneration 3 (2023) 217–231./ Engineered Regeneration 2022, 3 (1), 80-91/ Smart Medicine 2022, 1 (1), e20220006.// Smart Medicine 2022, 1 (1), e20220012.

Comments on the Quality of English Language

None

Round 2

Reviewer 2 Report

Comments and Suggestions for Authors

None